# A Data-Driven Noninteractive Authentication Scheme for the Internet of Vehicles in Mobile Heterogeneous Networks

**DOI:** 10.3390/s22228623

**Published:** 2022-11-09

**Authors:** Zongzheng Wang, Ping Dong, Yuyang Zhang, Hongke Zhang

**Affiliations:** School of Electronic and Information Engineering, Beijing Jiaotong University, Beijing 100044, China

**Keywords:** Internet of Vehicles, noninteractive authentication, network mobility, multihoming, multipath transmission

## Abstract

The rapid development of intelligent vehicle networking technology has posed new requirements for in-vehicle gateway authentication security in the heterogeneous Internet of Vehicles (IoV). The current research on network layer authentication mechanisms usually relies on PKI infrastructure and interactive key agreement protocols, which have poor support for mobile and multihomed devices. Due to bandwidth and interaction delay overheads, they are not suitable for heterogeneous IoV scenarios with network state fluctuations. In this study, we propose a data-driven noninteractive authentication scheme, a lightweight, stateless scheme supporting mobility and multihoming to meet the lightweight data security requirements of the IoV. Our scheme implements device authentication and noninteractive key agreement through context parameters during data communication. Due to saving the signaling interactive delay and certificate overhead, in the IoV scenario, the proposed scheme reduced the delay by 20.1% and 11.8%, respectively, in the authentication and handover processes and brought higher bandwidth aggregation efficiency.

## 1. Introduction

The Internet of Vehicles (IoV) is an essential component of the future Intelligent Transportation System (ITS) [1]. With the development of artificial intelligence applications represented by automatic driving, the in-vehicle sensors and user applications generate a large amount of multimedia data related to security issues [2]. The normal form of On-Board Units (OBUs) is in-vehicle gateways with a multihomed access capability, which can typically connect to multiple heterogeneous network resources, such as 4G/LTE, 5G, WiFi, and satellite links. As the core component of integrating heterogeneous network resources between Internet of Vehicles devices and cloud services in the ITS framework, the security of the in-vehicle gateway must be prioritized [3]. Multihoming and mobility cause frequent changes in the wireless network context of devices. To ensure the security and confidentiality of communication systems, we must create a gateway-to-cloud certified secure channel for mass data transmission [4]. Data loss caused by gateway devices during communication interruption caused by authentication switching should be prevented to meet the demand for delay-sensitive communication in the IoV. The unstable nature of a channel will lead to the loss of authentication messages, resulting in long-term communication interruption and a large amount of data loss. Therefore, higher requirements are placed on the authentication delay of equipment [5] to achieve context awareness. In summary, an appropriate authentication mechanism to meet the needs of the IoV scenario should be lightweight, low latency, and support both multihoming and mobility [6].

This study focuses on the security authentication of a mobile multihoming in-vehicle gateway in the IoV scenario, especially on the data source authentication scheme between the in-vehicle gateway and the cloud. How to realize in-vehicle gateway authentication in the dynamic context of the network and provide security-certified data support for IoV delay-sensitive applications is the purpose of our scheme. There are mainly three mainstream technologies to support mobility and multihoming in the IoV. The first is the network layer method of the Internet Key Exchange Protocol (IKE) and its extension (MOBIKE) [7]. The second is the Datagram Transport Layer Security (DTLS) [8] working in the transport layer. The last is the layer 3.5 Host Identity Protocol (HIP) [9] between the network layer and the transport layer. Its basic design allows mobility and multihoming. However, the core principle of the above research directions to construct the certification of devices remains the creation of Security Associations (SAs). The authentication of the in-vehicle gateway is achieved between two endpoints through the traditional Public Key Infrastructure (PKI) certificate mode. The key information maintained by SAs is realized through the interactive key agreement process with the Diffie–Hellman (DH) algorithm. In addition, IP handoff caused by mobility and multihoming support is realized through the interaction of control messages. In addition, since the original design did not consider the aggregation and utilization of heterogeneous network resources, the traditional security mechanisms could not achieve efficient aggregation of multiple links.

The role of the in-vehicle gateway is to provide data packet aggregation and forwarding. The SA establishes a connection between two communication parties, which violates the design principle of stateless forwarding of the network layer. Therefore, this SA-based security authentication measure is inefficient, expensive, and inflexible in a dynamic heterogeneous network scenario, which is intolerable for delay-sensitive IoV requirements.

The following list outlines the issues with the current mechanism:Identity authentication is dependent on the intricate PKI infrastructure. The operations related to certificates cause overhead issues in terms of delay and bandwidth.Control messages must be used to create and maintain the SA, which delays authentication and interferes with the IP layer’s stateless feature. Unavoidably, due to the link state fluctuation of heterogeneous networks, signaling interaction will bring unpredictable delay. At the same time, due to the unstable characteristics of the link, such as high packet loss, the performance of the mechanism is further deteriorated due to the retransmission of signaling.Existing research support for multihoming is restricted to only providing a list of alternate addresses to use multihoming interfaces as the primary and alternate channels. The bandwidth aggregation efficiency of multipath makes it difficult to reach the ideal value.

Given the above problems, we propose a data-driven noninteractive authentication scheme, a lightweight authentication scheme based on the network state context. It aims to realize data privacy and secure communication for the in-vehicle gateway with heterogeneous network access capability. Our scheme is based on bilinear pairing. The in-vehicle gateway calculates authentication keys from network state contexts, such as preassigned private keys, device IDs, network adapter serial numbers, and source address information. The initiation of device authentication and key agreement depends on the driver of user data rather than establishing a dedicated security state.

The security scenario we focus on is the data source authentication from the in-vehicle device with heterogeneous access capability to the cloud server. To secure the entire system, the in-vehicle device needs to achieve offline authentication and device ID registration in the cloud to establish an identity-based key system. The device ID, as a security credential, is used as an identity-based noninteractive key negotiation parameter in the data communication process. Combined with different access identifier parameters (usually IP addresses) and device context information, the multihomed device can authenticate the source address of multiple IPs during the mobile process. The cloud server can also ensure the consistency of data integrity and the data encryption key according to the device ID, access address, and other inherent information of the payload. The address source authentication and data confidentiality and integrity assurance during the communication process can be achieved through the above steps. The triggering of the security communication mechanism depends entirely on the interaction of the payload and does not need to establish the security state in advance. Compared with the traditional strategy, our scheme is more suitable for the heterogeneous Internet of Vehicles network with fluctuating network states due to its light weight and highly efficient characteristics. The specific authentication scheme is introduced in Section 3.

This article makes the following contributions.

A data-driven authentication scheme based on noninteractive key agreement is proposed. Compared with the PKI system based on certificate authentication, the proposed scheme has the characteristics of low bandwidth overhead and short interaction delay. This design paradigm meets the needs of the IoV scenario for data secure collection and analysis.According to the communication state of the in-vehicle gateway, we designed an authentication process and two handover mechanisms so that the proposed scheme inherits the stateless feature of the IP layer protocol. Furthermore, our scheme natively supports multipath.We proved the advantages of our scheme over the traditional interactive mechanism through a comparative simulation. In terms of the delay, it brings less authentication delay, handover overhead in different states, and calculation processing delay. In terms of multipath support, the proposed scheme has better heterogeneous resource aggregation capability.

The remainder is structured as follows. In Section 2, we introduce the relevant research. We outline our scheme’s precise design in Section 3. In Section 4, we conduct experiments to evaluate the performance of the suggested scheme in terms of the latency of authentication, handover, and data transmission. For the heterogeneous resource characteristics of the IoV, multipath aggregation efficiency is also an indicator that we focus on. Finally, we provide a summary in Section 5.

## 2. Related Work

### 2.1. IoV Security Architecture Based on Cloud Computing Scenarios

In the IoV, the in-vehicle gateway transmits data to the cloud through the insecure Internet. Security authentication is the main concern for insecure networks in terms of security. There has been much research on the security of the cloud architecture for the IoV. Reference [10] proposed a multifactor authentication mechanism in the cloud computing scenario. On this basis, Reference [11] designed a four-layer architecture based on a symmetric encryption algorithm and fog computing for the IoV authentication protocol. Given the dynamic nature and unreliable channels in the IoV, Reference [12] proposed a lightweight communication protocol for entities in the IoV. Reference [13] pointed out that the mechanism in [12] cannot resist key exposure, the man-in-the-middle attack, and realize mutual authentication between entities. They designed a secure and efficient information authentication protocol to resolve the security threats of the existing schemes. In order to reduce the cost of the security mechanism, Reference [14] designed a lightweight challenge corresponding authentication mechanism using the SDN architecture to overcome Denial-Of-Service (DOS), Address Resolution Protocol (ARP) spoofing, and sniffing in IoV scenarios.

The above research focused on the related security authentication process of the IoV in the cloud architecture scenario. The high dynamics of vehicles in IoV scenarios and the impact of complex multi-access heterogeneous resources on the security mechanisms were not paid attention to, and the characteristics of mobility and multihoming were not sufficiently discussed.

### 2.2. Key Exchange Protocol for Multihomed Mobile Devices

The elated research on security authentication and the key establishment process mainly focuses on improving the IKE, HIP, and DTLS.

The Internet Key Exchange Protocol (IKE) is a commonly used network layer authentication and key agreement mechanism. The Mobility and multihoming extension (MOBIKE) of IKE version 2 (IKEv2) is applied and studied in the Internet of Vehicles of heterogeneous networks. Figure 1 shows IKEv2’s initial exchange process. The certificate-related information required for authentication and the DH key negotiation process are completed in two rounds of interaction. For mobility and multihoming support, additional addresses are added to the optional support list. IKEv2 and IPsec were suggested as a security method by Fernandez et al. for IPv6 VANET in heterogeneous networks. Their strategy focused on securing IPv6 communication for internal vehicle equipment using different access interfaces, with wireless interfaces consisting of numerous heterogeneous connections. They mainly focused on mobility and seamless intertechnology handovers [15]. However, their multipath approach only supports the transmission of different streams and does not support parallel transmission. The authors of [16] suggested a lightweight IPSec-based authenticated key negotiation system for circumstances with limited resources.

HIP Diet EXchange (HIP-DEX) [17] is an improvement of the HIP specifically for resource-constrained devices, which reduces the complexity of cryptographic computation by removing digital signatures and encrypting session keys based on static elliptic curve Diffie–Hellman protocol, which maintains only the minimum security level. Figure 2 shows the HIP-DEX exchange process. Similar to the IKE, the HIP uses the DH algorithm with the puzzle mechanism to achieve key agreement. The exchange process also requires two rounds. Similarly, an address update notification is required for mobility and multihoming support. The C-HIP [18] is based on HIP-DEX and has an efficient key establishment component for highly resource-constrained IoT devices. The CHIP delegates expensive cryptographic operations to resource-rich devices in local networks. The implicit certificate-based mechanism was proposed for low-overhead demands in [19]. In [20], the authors presented sprayed strategies for hardening the better-than-nothing paradigm, minimizing the attack surface and, thus, the likelihood of an attack, with a focus on the opportunistic mode of the HIP in the IoT scene.

DTLS [8] is based on the Transport Layer Security (TLS) protocol using the User Datagram Protocol (UDP) for transporting services and provides equivalent security guarantees for end-to-end communication. The message flights for a full handshake are shown in Figure 3. The handshake of DTLS goes through three rounds of interaction. In the first round, the client sends a ClientHello message to start communication. The server verifies the reliability of the client by returning a verification request. The purpose of this step is to increase the cost of the client by maintaining a stateless cookie mechanism on the server, to fight against DDoS attacks to a certain extent. The next two rounds of handshake adopt similar mechanisms as IKEv2 and the HIP through certificate-based authentication and interactive DH key agreement algorithms. In [21], the authors demonstrated the availability of DTLS in an IoT network and proposed directions for improvement. In [22], the authors proposed a mobility-based DTLS authentication scheme to protect personal data collected and transmitted between mobile IoT devices and gateways. The scheme uses the session resumption function of the TLS protocol for authentication between mobile IoT devices and access gateways, while reducing the authentication overhead. In [23], the authors proposed CAT-Comp, a compressed sensing protocol that enables IoT devices to exchange compressed X.509 certificates and authorization tokens and send them over a low-power and lossy network, significantly reducing the payload size.

The above security authentication for mobile multihomed devices adopts PKI-based authentication and interactive key negotiation. The cost of certificates will cause bandwidth overhead. The interactive key negotiation has a failure probability due to packet loss in the scenario of link fluctuation.

Shamir introduced an Identity-Based Cryptographic (IBC) system in 1984. In this system, to reduce the expense of certificate authentication imposed by the conventional PKI system, the public key is retrieved by device identification [24]. In [25], the authors first proposed the Identity-based Noninteractive Key Distribution (ID-NIKD) scheme, which became an important reference for subsequent related research. The expansion of the security and the associated proof were carried out in [26]. Bilinear pairing on an elliptic curve provides the foundation for the scheme’s security. The IP identity signature authentication mechanism, which guards against IP spoofing attacks, was researched by [27]. Reference [28] studied the self-authentication mechanism of IP addresses in a LAN. A similar scheme was also used by [29]; noninteractive key agreement was used by the authors for vehicular communications.

We summarized and analyzed the existing research on identity-based noninteractive authentication mechanisms in the IoV. The existing mechanism relies on the Private Key Generator (PKG) registration of the fixed IoV component identification. Once the registration identification is determined, it cannot be modified. However, there is frequent switching of the access address during the moving process of the vehicle. Additional registration with the PKG is required for the source authentication of the switching address, which will increase the delay cost. Therefore, the authentication of the source address cannot be achieved. In addition, the existing research does not discuss the multihomed access capability of a single vehicle. Obviously, the above features are not fully applicable in IoV scenarios due to the mobility and multihoming features. Our security authentication architecture fully considers the above shortcomings; the device can realize source authentication for frequently switched multihomed access addresses through a single registration, so it has better mobility and multihomed support capability. Please refer to Table 1 for the schemes’ comparison.

## 3. The Design of Our Scheme

### 3.1. System Architecture

To realize the collection and transmission of massive sensor device data by intelligent applications, the multipath transmission technology integrating heterogeneous resources is indispensable in the IoV. The system architecture of multipath parallel transmission was based on our existing research [30]. In this multipath architecture, the in-vehicle gateway and cloud server are used as multipath aggregation agents, respectively, to realize multipath aggregation management by establishing logic tunnels to provide transparent concurrent multipath data transmission for massive sensor device data. The focus of this paper is on lightweight authentication support for mobile multihoming devices to achieve stateless forwarding of IoV data. Therefore, the comparison schemes in the subsequent experimental stages were all implemented under the same architecture. It is worth noting that the schemes proposed in this paper can be integrated at different levels according to specific requirements and can realize IP-in-IP encapsulation at the network layer or UDP encapsulation at the transport layer according to the mode adopted by the architecture. In order to facilitate the comparison with the subsequent experimental comparison scheme in this paper, we integrated the proposed scheme into the network layer through IP-in-IP encapsulation. Figure 4 shows our system architecture. Different in-vehicle gateways can communicate with cloud servers through heterogeneous networks. We show the registration and communication process of building security domains with different GWs under the same CS. Table 2 shows the abbreviations in our paper.

### 3.2. Security Assumptions

For the convenience of illustration, we use the abbreviations GW and CS to represent the in-vehicle Gateway and Cloud Server, respectively. The following security presumptions form the foundation of our security system. The GW is a trusted device and will not be an attacker against system security. When the GW is hijacked by an attacker and becomes no longer trusted, the system will ensure system security through key revocation.

We designed an identity-based noninteractive key agreement to construct the security domain. The theory is shown below.

Pairings and the Bilinear Diffie–Hellman (BDH) problem:

$G_{1}$ represents an additive group of prime order *q* in a bilinear group system. $G_{T}$ represents a multiplicative group of prime order *q* that we define. Based on the above definitions, $e:G_{1}\times G_{1}\to G_{T}$ stands for a bilinear map. The bilinear map has the properties listed below:

(a) Bilinearity: ∀P,Q∈$G_{1}$ and ∀a,b∈$Z_{q}$, we have $e\left(aP,bQ\right)=e{\left(P,Q\right)}^{ab}$.

(b) Non-degeneracy: e(P,P) is a $G_{T}$ generator if *P* is a $G_{1}$ generator.

(c) Computability: e(P,Q) can be efficiently calculated given *P*∈$G_{1}$ and *Q*∈$G_{1}$.

The BDH problem determines the security of the key negotiation procedure depicted below.

Given the above-mentioned generation algorithm, PairingGeneration, we describe the advantage of a probabilistic polynomial-time method adv in addressing the BDH issue as
(1)$${\mathrm{Adv}}_{𝒜}^{\mathrm{bdh}}=Pr(𝒜\left(aP,bP,cP\right)=e{\left(P,P\right)}^{abc})$$
where $a,b,c\leftarrow Z_{q}^{*}$ are picked at random in a uniform manner. The BDH assumption holds for ($G_{1}$,$G_{T},e$) if ${\mathrm{Adv}}_{𝒜}^{\mathrm{bdh}}$ is negligible for all probabilistic polynomial time algorithms adv.

### 3.3. System Establishment

**System setup**: The CS generates a bilinear group system using a probabilistic polynomial-time algorithm, which includes two-cycle groups $G_{1}$ and $G_{T}$ with prime order *q*, a bilinear map $e:G_{1}\times G_{1}\to G_{T}$. The CS publishes the hash function $H:{0,1}^{*}\to G_{1}$, which is collision-resistant according to the security requirements. The devices in the system must be uniquely identified, as in $ID_{CS}$, $ID_{GW}$. The bilinear system created by the current CS is the foundation upon which the security domain is built. The system parameters $<G_{1},G_{T},P,e,q,H>$ published by the CS ensure that the operations between devices belonging to the CS are based on the same system. We follow the entities of the bilinear system parameter operations to form the public security domain.

### 3.4. Establish the Security Domain of GW

**Registration phase**: Different from the traditional IBC system, in the proposed security scheme, the CS, as the communication participant of the security domain, only provides the public parameters of the security domain and the CS device ID as the communication party. The CS does not generate the private key in the domain. The private key of the GW device does not need to be calculated by the CS. Unlike the system that relies on the PKG, it provides the key certificate to the CS by registering the GW device with the CS. This ensures source address authentication when the GW device’s IP address and other key negotiation parameters are variable.

In the registration process, the GW device generates the private key $s_{GW}\in$$Z_{q}$ by itself and generates the registration key $S_{CS}^{GW}$ through the scalar multiplication operation on the elliptic curve according to Equation (Equation 2). We concatenated the $ID_{CS}$ and $IP_{CS}$ strings and mapped them to the points on the elliptic curve through the H() function. The result performs scalar multiplication on elliptic curves with $s_{GW}$ to obtain $S_{CS}^{GW}$. The GW submits the registration key $S_{CS}^{GW}$ to the CS. The CS stores the accompanying private key $S_{CS}^{GW}$. In the subsequent key negotiation process, the CS will calculate the real-time shared key based on the security domain registration key and the communication context of both communication parties, so as to make the device’s IP address variable. Address authentication based on the device communication context was implemented.
(2)$$S_{CS}^{GW}=s_{GW}H(ID_{CS}||IP_{CS})$$

Figure 5 depicts the relevant process. We might consider the CS and GW to be part of the same security domain. The registration process’s security parameters make up the security domain’s context parameters. Through the registration phase, the CS and a GW share the key negotiation parameters based on the master private key in the public security domain. Because the CS has the registration credential, it can ensure that the key negotiation with different parameter IDs of the GW can be realized. The CS and the registered GW jointly constitute a sub-security domain.

### 3.5. Authentication and Key Agreement

As was already noted, the creation of the SA is necessary for the current network layer security approach to operate. In wireless heterogeneous settings, there is a large delay introduced because state establishment information exchange is necessary prior to data interaction between communication parties. The foundation of our suggested security system is a context-aware data-driven stateless protocol. It is not necessary for both communication parties to initially establish a stateful connection when both have communication requirements. In this work, the secure authentication of multihoming mobile devices was accomplished via an identity-based noninteractive key. In this arrangement, the security domain includes both the GW and CS. Key negotiation was carried out between the two communicating parties utilizing security domain parameters and network state context parameters. The support for an address change is made possible by this design.

After the registration phase, a security domain database is established on both sides of the GW and CS to manage the network context status parameters under the same security domain for data authentication. Figure 6 shows the context parameter status when the same GW and different CSs constitute a security domain. The security domain master key $s_{GW}$ and the CS device’s ID are recorded at the GW end. Each GW ID and its corresponding $S_{CS}^{GW}$ are recorded at the CS end. At the same time, the network card serial number and the IP address from the corresponding device, as well as the switching times of the corresponding IP address are recorded. These are used as the network context parameter to generate the authentication key. At the same time, it records the IP switching frequency and packet transmission characteristics of the corresponding network card of the corresponding device. These are used as one of the authentication parameters for a more advanced security access design. The CS side’s context parameter database is shown in the figure. The device ID parameter indicates the parameters in the security domain to which both sides of the communication belong. The IP address is used to calculate the materials of the shared key to realize the source address authentication. The network card serial number and IP handover times are used to characterize the scheduling characteristics and mobile handover characteristics of the mobile multihoming device. Both sides of the communication maintain the switching times. Each time the IP address side from the same network card serial number of the GW triggers the count +1 at the CS end, both sides of the communication have the same switching count.

Figure 7 shows the authentication and data transmission process, and the relevant description is as follows.

#### 3.5.1. GW End

When the GW needs to send data, it dynamically senses the network status, extracts the network adapter information, and obtains the address and network adapter serial number $IF_{num}$. The private key $S_{GW}^{IF_{num}}$ for the network interface $IF_{num}$ is derived from Equation (Equation 3). We concatenated the $ID_{GW}$, $IP_{GW}^{IF_{num}}$, and $IF_{num}$ strings and mapped them to the points on the elliptic curve through the H() function. The result performs scalar multiplication on the elliptic curves with $s_{GW}$ to obtain the private key $S_{GW}^{IF_{num}}$.
(3)$$S_{GW}^{IF_{num}}=s_{GW}H(ID_{GW}||IP_{GW}^{IF_{num}}||IF_{num})$$

After obtaining the private key of the network interface, the shared key material can be determined. The security domain information and destination IP address of the communication party are necessary. The key material $K_{GW}^{IF_{num}}$ is derived through Equation (Equation 4). In this process, $S_{GW}^{IF_{num}}$ and $H(ID_{CS}||IP_{CS})$ are paired in the bilinear system of the current security domain.
(4)$$K_{GW}^{IF_{num}}=e(S_{GW}^{IF_{num}},H(ID_{CS}||IP_{CS})$$

The authentication key is calculated by the Pseudo-Random Function (PRF) with input parameters $K_{GW}^{IF_{num}}$ and $seed_{initial}^{IF_{num}}$, as shown in Equation (Equation 5).
(5)$$k^{IF_{num}}=PRF\left(K_{GW}^{IF_{num}},seed_{initial}^{IF_{num}}\right)$$

The number of IP address switches of the network adapter is expressed as $seed_{initial}^{IF_{num}}$. Each network card’s seed counter is initially set to 1 or a default initial value. As the network adapter’s IP changes, the counter step size also changes. To guarantee the consistency of shared keys, both communication participants must keep the same seed counter. To assess the safety status of the equipment and the safety factor of the equipment operating cycle, this parameter can be utilized as a context-aware parameter input.

The network adapter identifying the number of the local device is a necessary context parameter in the multihoming scenario, which aids in the management of the multihoming address. Additionally, the GW ID needs to be transmitted to the other end, in order to determine the necessary security domain characteristics. In Figure 8, the packet header is displayed. By performing the HMAC procedure on the IP packet’s immutable portion, the authentication field’s value may be determined. To guarantee data integrity and source authentication, the HMAC key is the agreed-upon shared key.

#### 3.5.2. CS End

When the CS receives a data packet, it parses the packet header information and searches the security domain database for the relevant item. When the corresponding item exists, the peer device is legal. If the packet from the peer network adapter that has not been initialized is received, the relevant shared key material is initialized according to Equation (Equation 6).
(6)$$K_{GW}^{IF_{num}}=e(H(ID_{GW}||IP_{GW}^{IF_{num}}||IF_{num}),S_{CS}^{GW})$$

In order to verify the integrity of the received packet and realize its source authentication, the shared private key is then retrieved by the PRF. The CS uses the authentication key to perform the same HMAC calculation with the sender to compare the authentication fields of the packets. The entire communication flow is depicted in Figure 9.

Obviously, just a few fields must be added to the data packet header in order to accommodate the authentication information stated above. Without requiring any further information exchange, the authentication procedure may be started when a user data transmission request occurs on the mobile subnet. It exactly adheres to the IP layer’s design philosophy, whereby only forwarding is implemented, and the reliability of the communication is ensured through the top layer.

From the above process, we can see that there is only normal load data exchange in the whole authentication process, and there is no additional signaling transmission. In the IOV scenario, various devices generate new data all the time. This scheme avoids the loss of data blocking caused by the time loss in the signaling interaction process.

### 3.6. Mobile Support

When IP address switching occurs, the network adapter of the GW dynamically detects the change of the address context and dynamically updates the new private key of the corresponding network interface as Equation (Equation 7).
(7)$$newS_{GW}^{IF_{num}}=s_{GW}H(ID_{GW}||newIP_{GW}^{IF_{num}}||IF_{num})$$

The shared key material should be updated as Equation (Equation 8).
(8)$$newK_{GW}^{IF_{num}}=e(newS_{GW}^{IF_{num}},H(ID_{CS}||IP_{CS}))$$

When calculating the shared key using the PRF as Equation (Equation 9),
(9)$$newk^{IF_{num}}=PRF\left(newK_{GW}^{IF_{num}},seed_{initial}^{IF_{num}}+1\right)$$

The seed counter is incremented by one with the IP switching, forming a new shared key. Then, it communicates as described above.

The receiver receives the data packet from the new address. The address update can be detected by evaluating the network card serial number in the header, and the shared key material is recalculated as Equation (Equation 10).
(10)$$newK_{GW}^{IF_{num}}=e(H(ID_{GW}||newIP_{GW}^{IF_{num}}||IF_{num}),S_{CS}^{GW})$$

In addition, the PRF will increase the seed counter by one when calculating the shared key. If the updated packet fails to pass HMAC authentication, it is a forged attack packet, and thus, the corresponding shared key and seed counter will not be updated.

Since the proposed scheme is triggered by the transmission of user data, there are different authentication switching modes depending on whether there is data transmission between the GW and CS. In the first scenario, switching takes place while both sides are exchanging data. Instead of being provided independently, the IP switching information is delayed until the user data packet is updated and activated. This mode is named the transmission mode. The second case is that there is no data transmission between the GW and CS. The GW will now aggressively transmit alerts of address updates to notify the CS. This mode is named the notification mode. According to the above analysis, no matter in which mode, the switching process only needs a single information interaction at most to achieve address and key renegotiation. Our scheme avoids the data transmission interruption caused by redundant message interaction in traditional stateful protocols.

### 3.7. Key Revocation

In our design, the device registration process is realized through a secure channel. That is, the device is a secure device that passes the review when it is registered. In addition, each device holds the independent device key $s_{G}W$ in the communication process. Even if a single device is attacked, it cannot affect the security of other devices in the system. Since the CS maintains the GW communication context information in the security domain, when the GW becomes a malicious device, the CS can identify the malicious device and revoke its key according to abnormal changes in the context, such as frequent switching of the address, which causes the $seed^{IF_{num}}$ changes and the abnormal transmission of data streams. In addition, the design of key revocation can also enhance the robustness of the system against malicious devices’ attacks.

Under our security domain architecture design, the key revocation is straightforward. The CS only needs to publish the revocation list of the GW equipment and broadcast it in the whole domain. For scenarios with high security requirements, the characteristics of the IBE technology can be used to add a time parameter when generating the registration certificate when the GW registers with the CS according to the requirements of the key security period. In this way, the key expiration cancellation can be realized. The above two schemes can be used in combination. Through this design, our system security is guaranteed in terms of the timeliness and security level.

### 3.8. Source Address Authentication against Man-in-the-Middle Attack

Our scheme implements the authentication of the communication source address based on the device ID and communication context. We explain this through the following description. Suppose an attacker impersonates the identity of a legitimate device $ID_{GW}$ with $IP_{Adv}$ to communicate with the CS. The CS calculates the authentication key according to equation $K_{GW}^{IF_{num}}=e(H(ID_{GW}||IP_{Adv}^{IF_{num}}||IF_{num}),S_{CS}^{GW})$. An attacker who wants to pass the message authentication needs to generate an authentication header through the authentication key. In this way, the CS can conclude that the device with $IP_{Adv}$ is an attacker, thus denying the address access. Any changes made by an attacker in the communication process will make the other legal party unable to generate the authentication header corresponding to the packet context. Such changes will only lead to invalid data and cannot lead to key disclosure. The data flow that has not been attacked can still generate the corresponding security key through a legitimate process to achieve secure communication. In addition, because of our source address authentication function, attackers cannot perform man-in-the-middle-attacks as devices counterfeiting any communication.

### 3.9. Computational Overhead

We present the computational cost for all cryptographic operations in our scheme in Table 3. The configuration of the test equipment was “Ubuntu 20.04, with an Intel (R) Core (TM) CPU i5-7200U,8.1 GB memory, @2.50 GHz”. The source code was implemented in C++.

The overhead of system establishment can be loaded when the device starts running, including $T_{ini}$ at the GW end and $T_{ini}+T_{convert}$ at the CS end. In the authentication phase, the total computational cost of our scheme is $3T_{convert}+T_{multi}+2T_{map}+2T_{auth}=8.194$ ms. It should be noted that our scheme can greatly reduce the delay in the interaction process. In the context of the IoV with fluctuating link states, the delay occupies a more important proportion than the computational cost.

## 4. The Experimental Verification of the Proposed Scheme

In this part, we compare the proposed scheme with the interactive authentication mechanism in four aspects: authentication latency, handover latency, data latency, and multipath aggregation efficiency, so as to evaluate the performance of the proposed scheme:Authentication latency is the time cost between the GW starting the authentication process and finishing the shared key agreement.The time cost between the GW reacquiring the address, updating the binding with the CS, and re-negotiating the shared key is defined as handover latency.Data latency is the time cost caused by the calculation and processing of the schemeMultipath aggregation efficiency is the effect of the security scheme on bandwidth aggregation in heterogeneous networks.

The authentication and handover latency is an important indicator related to traffic safety for highly delay-sensitive IoV applications. Due to the impact of vehicle mobility and network heterogeneity, the packet loss rate and the delay of the link will greatly affect the authentication and handover process. The latency indicator evaluates the most-important impact of the security scheme in IoV applications. The ideal security mechanism should be able to efficiently complete authentication in the case of poor link status. The data transmission delay reflects the complexity of the security scheme to a certain extent, and the gap between different security schemes is mainly reflected in the calculation and processing delay. With the development of multimedia services in the IoV, the demand for network bandwidth is growing. The bandwidth aggregation efficiency reflects the impact of the security scheme on network transmission efficiency when the network load increases.

According to the related work, we selected the mechanism in [15] as the representative of the interactive mechanism from the existing research. This work considered the authentication research in the mobile multihomed scenario, which is consistent with our research direction. For the noninteractive mechanism based on bilinear pairs, we selected the relevant latest research [29] as a comparison. However, it does not involve the research on handover and multihomed support. According to its idea, we integrated it into our simulation environment to compare the authentication delay. The interactive scheme was expressed as INTERAC and the noninteractive scheme was named as SFVEC in [29].

### 4.1. Network Topology for Simulation

To simulate mobile heterogeneous network scenarios, we designed a network topology, as shown in Figure 10. The link parameters of three links were designed to simulate heterogeneous network scenarios, and the mobile switching scenarios were simulated through the GW switching at different access points. We constructed a multipath aggregation channel by building a virtual tunnel between the multi-interface GW and CS. The implementation of multipath transmission was realized through the Netfilter mechanism of the Linux kernel. We implemented packet interception and multipath scheduling by registering HOOK functions. The proposed authentication scheme uses the netlink mechanism to calculate and update the authentication key stored in the kernel. The scheduling policy was set to ECF by default. The Pairing-Based Cryptography (PBC) library was used to implement our security method. The pseudo-code of the scheme proposed is referred to Algorithms 1 and 2.
Figure 10Simulation of the mobile heterogeneous networks through the design of link parameters and IP switching.
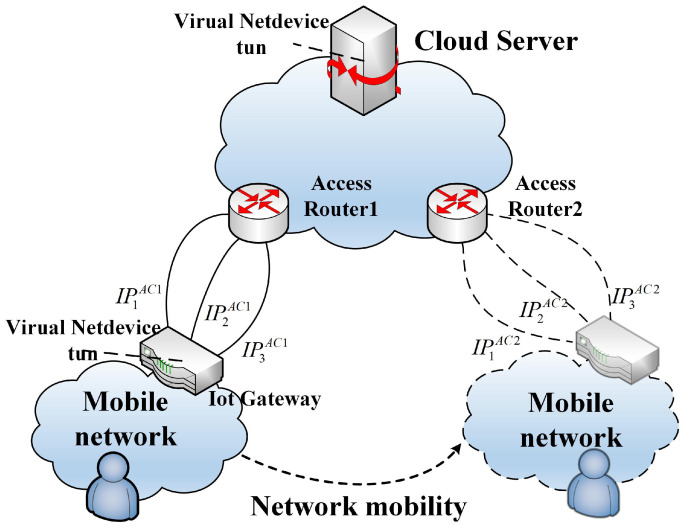

**Algorithm 1** GW end.1:**function**Config_Initialization()2:      Establishment (pairing, $s_{GW}$)3:      Global.Pk_CS($IP_{CS}$, $ID_{CS}$)4:**end function**5:**function**Network_management ()6:      **while** true **do**7:            **if** linkupdateflag==true **then**8:                 Uplink.Sk_GW(serialno, $IP_{GW}$, $ID_{GW}$)9:                 Uplink.sharekey(SkGW, Global.Pk_CS)10:           **end if**11:     **end while**12:**end function**13:**function**Data_processing ()14:      **while** recv(GW_tun) **do**15:            Path_Selection()16:            Uplink.auth_header()17:            Uplink.data_send()18:      **end while**19:**end function**

**Algorithm 2** CS end.
1:**function**Config_Initialization ()2:      Establishment (pairing, $s_{GW}$)3:
**end function**
4:**function**Data_processing ()5:      **while** recv(CS_Uplink) **do**6:            Header_parsing()7:            **if** header.device_ID==LICENSED **then**8:                  **if** header.device_ID==CONFIG **then**9:                        **if** header.Interface_no==CONFIG **then**10:                            **if** header.ADDRESS==ORIGINAL_ADDRESS **then**11:                                  **if** Uplink.auth()==1 **then**12:                                        Date.forward()13:                                  **else**14:                                        Date.discard()15:                                  **end if**16:                            **else**17:                                  Uplink.sharekey($newIP_{GW}$)18:                            **end if**19:                      **else**20:                            Uplink.loadserialno($IP_{GW}$, serialno)21:                      **end if**22:                **else**23:                      Uplink.loadGW($IP_{GW}$, $ID_{GW}$, serialno)24:                **end if**25:          **else**26:                Data.discard()27:           **end if**28:      **end while**29:
**end function**



In the program loading phase, both the CS and GW run the CONFIG_INITIALIZATION () function to load the parameters required for the construction of the bilinear system, that is to complete the loading of the parameters $<G_{1},G_{T},P,e,q,H>$, so that the security mechanism runs on the bilinear system with the same parameters.

At the GW end, the NETWORK_MANAGEMENT () thread detects the access status of the network adapter through the trigger mechanism, obtains the adapter information, including the serial number of the network adapter and the IP address, and calculates the corresponding private key through the bilinear operation combined with its own private key, SKGW. Thread DATA_PROCESSING processes the user packet to be transmitted. Firstly, the multipath scheduling function Path_Selection () selects the path to send. auth_header () calculates the corresponding key according to the selected network card to generate the authentication header. Finally, data_send () implements the construction and sending of the authentication packets.

At the CS end, the DATA_PROCESSING () thread calls the Header_parsing () function to parse the packet, checks the registration through the device ID, and checks the private key of the corresponding network adapter. If the packet of the corresponding network adapter is received for the first time, the corresponding key is generated according to the packet parsing parameters, through the auth () function, which checks the HMAC result of the packet header. If it passes the authentication, it receives the packet. Otherwise, it discards the packet.

In the process of our experimental setup, we set the parameters according to our team’s long-term research in the heterogeneous Internet of Vehicles. For the test research of the relevant realistic parameters, please refer to [31]. In the experiment, we selected the range of delay and the packet loss rate under real conditions. The experimental results are of practical significance. In the selection of the authentication and handover delay, our delay range was 0-100ms. The setting of the packet loss rate fully considered the scenario of an extremely bad link environment. Table 4 shows the range of the heterogeneous parameters under the general conditions of vehicle networks.

### 4.2. Authentication Latency

In order to fully evaluate the authentication latency of the proposed scheme in the heterogeneous network scenario of mobile multihoming devices, we replicated the high packet loss rate and significant link state variation features of mobile heterogeneous networks by adjusting the connection latency under various packet loss rate scenarios. From the results shown in Figure 11, we concluded that, under the same loss rate, the authentication delay overhead of our scheme had a smaller slope than the interactive scheme. Under the same link delay, our scheme also had less delay overhead under different packet loss rates, and this advantage became more obvious as the link deteriorated. The delay reduction by the proposed scheme was attributed to two factors. Firstly, for the scenario with the same packet loss rate, compared with the two or more rounds of information interaction required by the interactive authentication mode, the noninteractive authentication feature of the proposed scheme resulted in a minor authentication delay. It increased positively with the increase of the link latency, but the interactive authentication overhead had a higher growth rate than the proposed scheme. Secondly, when considering the fixed delay and the change of the packet loss rate, with the increase of mutual information, the certificate and other schemes adopted in the interactive mode had higher bandwidth overhead, which led to a greater possibility of packet loss and a higher probability of packet loss. In order to establish a stateful security alliance, more timeout retransmission mechanisms will be triggered, resulting in a significant increase in overhead. However, the proposed scheme is a stateless mechanism triggered by packet forwarding. It will not cause the authentication establishment process to block normal data transmission.

### 4.3. Handover Latency

This study was solely concerned with the IP address swapping caused by mobility. In this part, we compared the MOBIKE extension of IKEV2 with the two switching modes of the proposed scheme in terms of handover delay performance. Figure 12 shows the results. Since there was no interactive state transfer process, the transmission mode preserved the stateless properties of IP, thus reducing the switching overhead. At the same time, in the notification mode, compared with interactive handover, the proposed scheme uses noninteractive key agreement to reduce the number of interactive rounds, thus reducing the handover delay.

From the results shown in Figure 12, we can see that, in the transmission mode, the performance of the handover delay of the proposed scheme was similar to that of the authentication delay shown in Figure 11. This is because the noninteractive characteristics make the heavy key negotiation information required for handover be carried in the normal load communication process. Compared with the interactive mechanism, the amount of information required is greatly reduced, thus reducing the delay overhead, even in the notification mode. Our scheme had fewer interaction times than the interactive security federation state switching, and the delay overhead also had a smaller growth slope.

We provide statistics on the above authentication delay and handover delay data. Table 4 shows the performance improvement in different scenarios. The numbers in the table represent the average latency reduction percentage brought by each process under different packet loss rate settings, and the entries represent two schemes for comparison.

### 4.4. Data Latency

In this part, we evaluate the calculation and processing costs of the proposed scheme. The proposed scheme and IKEv2 AH protocol used the SHA256 HMAC algorithm for comparison. Table 5 displays the multilink state parameters used to simulate the heterogeneous multilink scenario.

We calculated the RTT delay of data by sending ping packets to the opposite CS. We compared three scenarios: using the proposed scheme, establishing the SA for the corresponding link in IKEv2 mode, and no authentication. The results are shown in Figure 13. We can draw the conclusion that, in the case that the heterogeneous link characteristics of multipath will magnify the impact of computing and processing overhead on data transmission delay, the authentication mechanisms will bring different degrees of data latency, but the proposed scheme has less impact and latency fluctuation. According to the data statistics in Figure 13, under the link setting as Table 5, the data latency of the proposed mechanism was reduced by 4.17% compared with the other schemes.

To evaluate the overall delay advantage of the scheme, we conducted experiments in the IoV scenario with 1Hz as the data acquisition frequency according to the environmental settings in Table 5. The proposed authentication scheme can effectively reduce the authentication delay and handover delay. Under the above environmental settings, the delay was reduced by 20.1% and 11.8%, respectively. More improvement will be brought if the data acquisition frequency increases.

### 4.5. Throughput

Finally, we evaluated the multipath aggregation efficiency of the proposed scheme. The significance of the multipath parallel transmission system is that it achieves the aggregation of heterogeneous resources and improves the transmission bandwidth. Therefore, the ideal security scheme should try to ensure aggregation efficiency under the premise of security. Bandwidth aggregation efficiency is greatly affected by the link state and scheduling strategy. In order to evaluate the aggregation efficiency of the security mechanism, we set the heterogeneous links to fluctuate between different link characteristics and adopted the theoretically optimal scheduling algorithm ECF. In order to compare the impact of the security mechanism on the system aggregation efficiency, we evaluated the aggregation efficiency under the maximum load. The heterogeneous link parameter settings are shown in Table 5. We compared the UDP maximum theoretical bandwidth transmission between the GW and CS through iperf. The experimental results are shown in Figure 14. The experimental results showed that the proposed scheme can bring higher bandwidth aggregation efficiency under the same load and scheduling strategy. According to the data statistics in Figure 14, the throughput of the proposed scheme increased by 4.34% compared with the other scheme. The reason for this advantage is that the proposed scheme described in Section 3 has a lower transmission delay. The lower transmission delay can better realize the aggregation of data packets in delay-based scheduling mechanisms such as ECF. In addition, in order to further analyze the reasons for the improvement of bandwidth efficiency, we calculated the phenomenon of packet disorder during transmission. The statistics of out-of-order packets are shown in Figure 15. The results showed that the proposed scheme had fewer out-of-order packets, which improved the bandwidth aggregation efficiency of the scheduling strategy.

## 5. Conclusions

In this paper, we focused on the highly dynamic and heterogeneous characteristics of the IoV and proposed a lightweight security authentication scheme to support mobile and multihomed in-vehicle devices. Our scheme adopted the identity-based noninteractive authentication and key agreement mechanism. The authentication and handover renegotiation were triggered by the user payload, preserving the stateless nature of the network layer. Identity-based and noninteractive features made up for the high certificate cost and interactive delay introduced by the widely used interactive authentication scheme. Extensive simulations were carried out for comparison with other interactive schemes to prove the superiority of the proposed scheme in terms of latency and multipath aggregation efficiency in different network states. The results showed that our scheme had better performance in the heterogeneous-resource IoV scenarios with frequent link fluctuations. The emphasis of the scheme proposed in this paper was to improve the flexibility of the authentication mechanism by reducing the interaction delay. The expansion of the system scale and the intrusion detection of abnormal devices will be our essential research directions in the future.

## Figures and Tables

**Figure 1 sensors-22-08623-f001:**
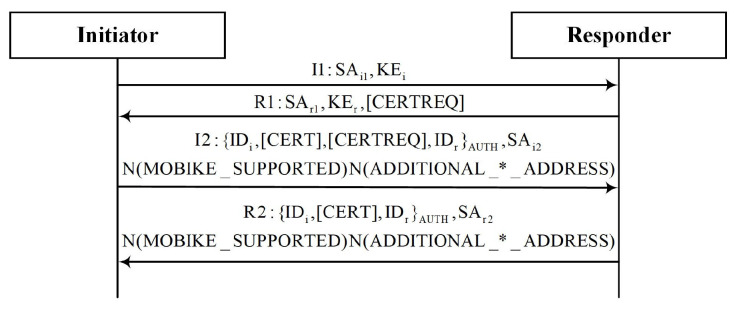
IKEv2’s initial exchanges.

**Figure 2 sensors-22-08623-f002:**
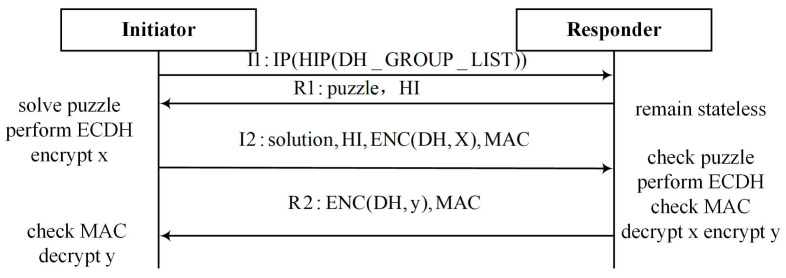
HIP-DEX exchange.

**Figure 3 sensors-22-08623-f003:**
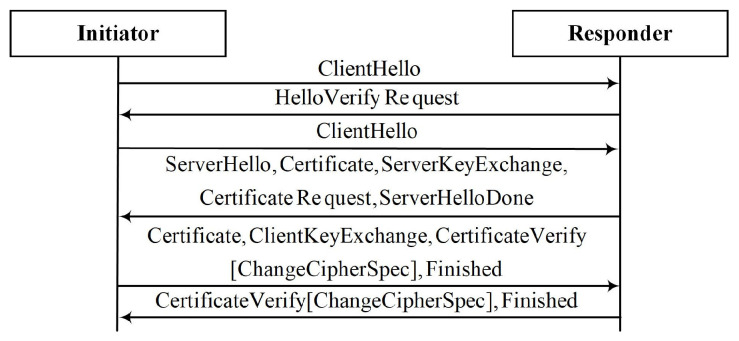
DTLS base exchange.

**Figure 4 sensors-22-08623-f004:**
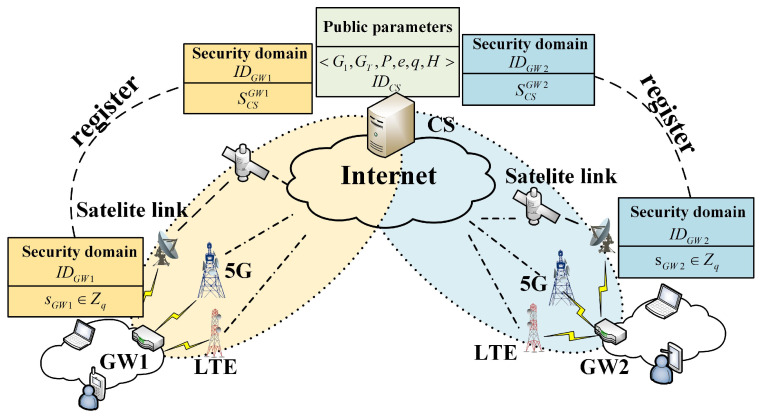
The system architecture of the registration, authentication, and communication process of the security domain composed of multiple devices; mobile GWs in the architecture can access a heterogeneous network.

**Figure 5 sensors-22-08623-f005:**
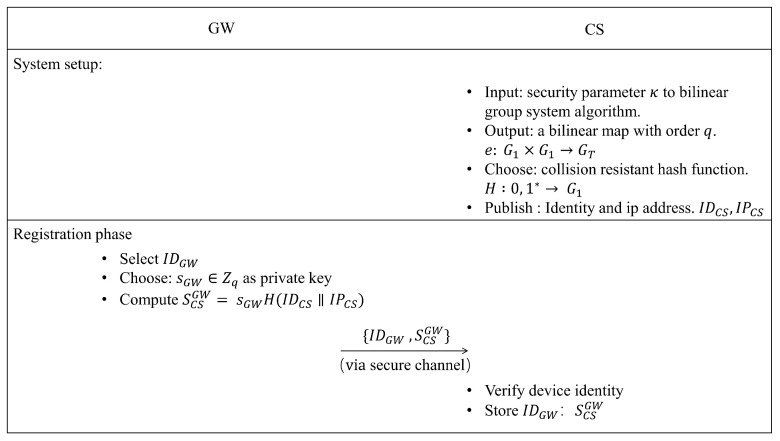
System setup and registration process.

**Figure 6 sensors-22-08623-f006:**
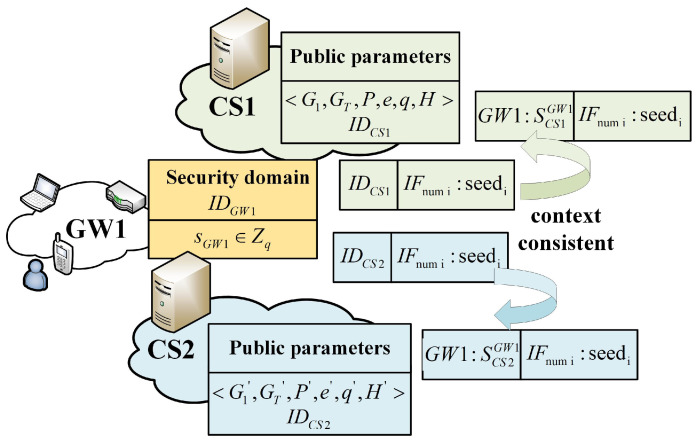
Context parameter status of the same device in different security domains.

**Figure 7 sensors-22-08623-f007:**
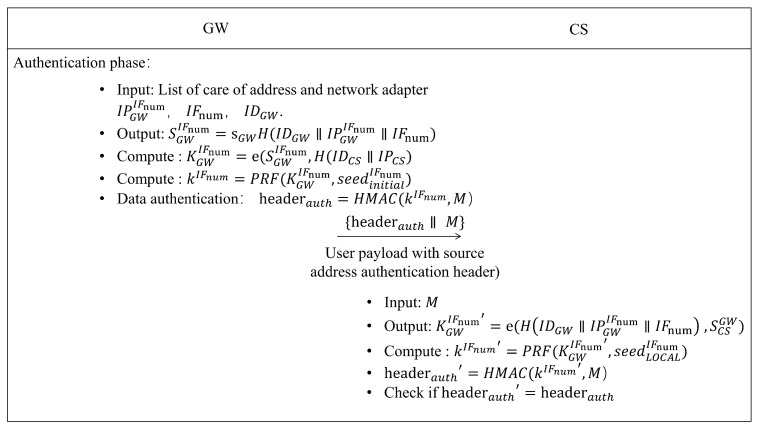
Authentication and data transmission process.

**Figure 8 sensors-22-08623-f008:**
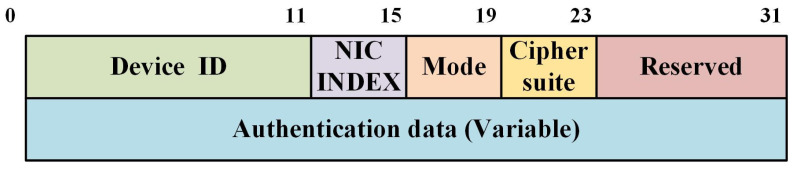
Packet header.

**Figure 9 sensors-22-08623-f009:**
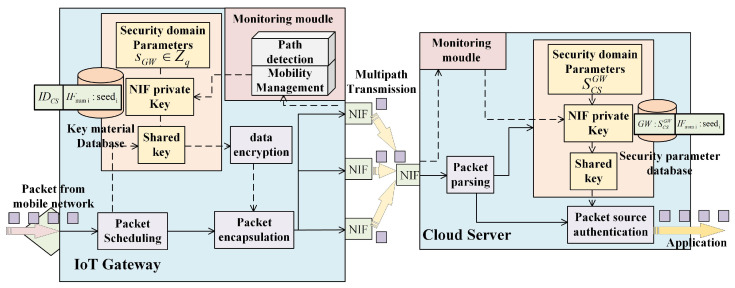
Function module design of the GW and CS in the communication process, presenting the processes of context monitoring, key generation, and data communication.

**Figure 11 sensors-22-08623-f011:**
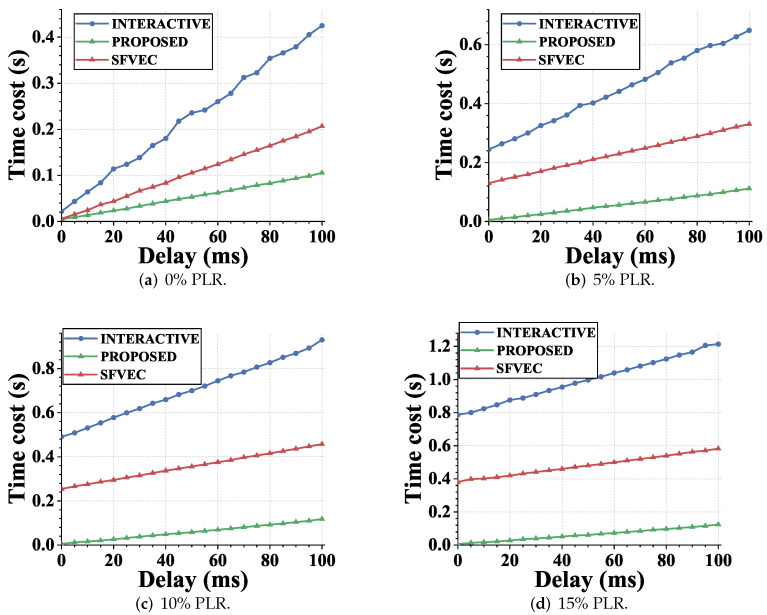
Authentication latency results under different link settings. The abscissa represents different link delay settings, and the subgraph represents different link packet loss rate settings. Authentication latency is defined as the time cost between the GW starting the authentication process and the shared key being negotiated. (**a**) Packet loss rate = 0%, (**b**) packet loss rate = 5%, (**c**) packet loss rate = 10%, and (**d**) packet loss rate = 15%.

**Figure 12 sensors-22-08623-f012:**
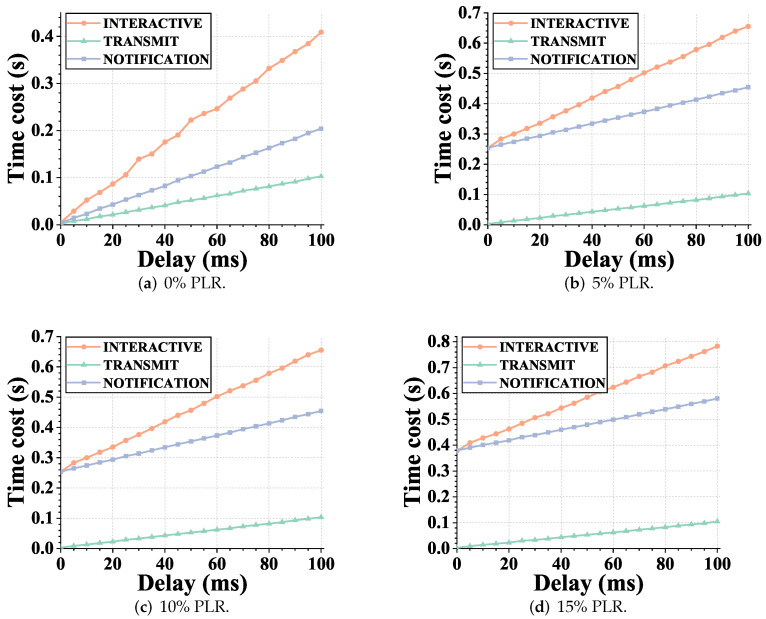
The handover latency results under different link settings. The abscissa represents different link delay settings, and the subgraph represents different link packet loss rate settings. Handover latency is defined as the time cost between the GW reacquiring the address, updating the binding with the CS, and re-negotiating the shared key defined as the handover latency. (**a**) Packet loss rate = 0%, (**b**) packet loss rate = 5%, (**c**) packet loss rate = 10%, and (**d**) packet loss rate = 15%.

**Figure 13 sensors-22-08623-f013:**
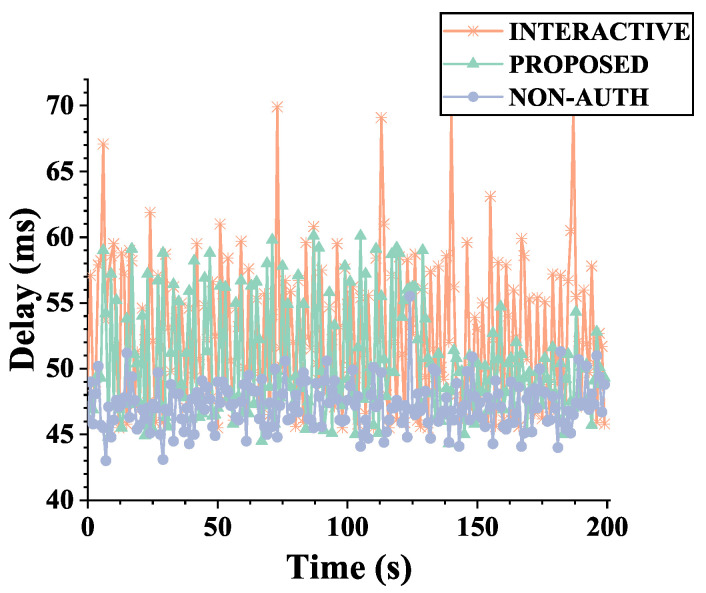
Data transmission delay under multipath settings; comparison of the proposed scheme, interactive scheme, and no security scheme.

**Figure 14 sensors-22-08623-f014:**
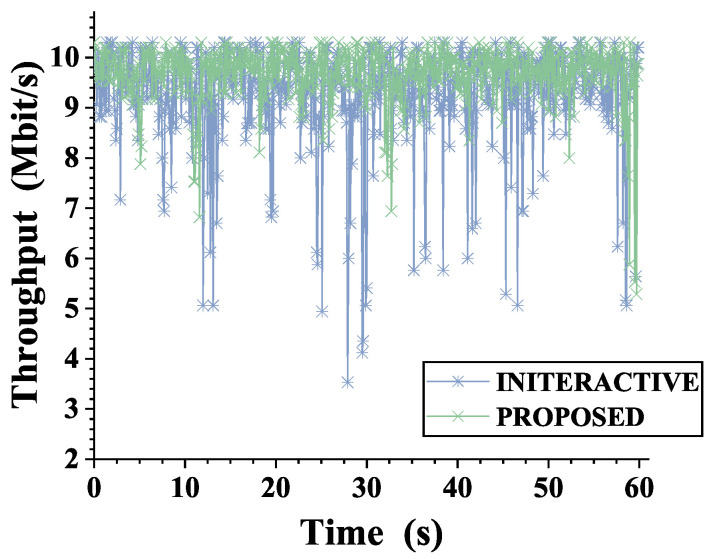
Comparison of bandwidth aggregation efficiency between the proposed scheme and the interactive scheme under multipath settings.

**Figure 15 sensors-22-08623-f015:**
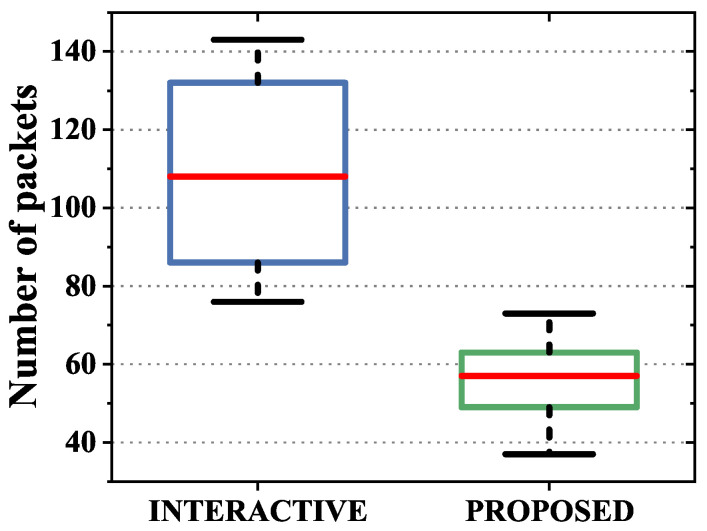
Box plot statistics of out-of-order packets of the proposed scheme and interactive scheme.

**Table 1 sensors-22-08623-t001:** Comparison of existing studies.

	Expansion	Layer	Handoff Support	Multihoming Support	Lightweight
IKEv2	[15,16]	Network layer	SA status needs to be updated by notification (MOBIKE extension)	Additional address as alternate path (MOBIKE extension)	✕
HIP	[17,18,19,20]	Between transport and network layer	LOCATOR status needs to be updated by notification	Only backup mode is supported	✕
DTLS	[21,22,23]	Transport layer	Application layer reconnection is required	No related extensions	✕
IBC	[27,28,29]	Application layer	Address representation change is not supported	No related extensions	√

**Table 2 sensors-22-08623-t002:** Summary of symbols.

Symbols	Description
IKE	Internet Key Exchange Protocol
HIP	Host Identity Protocol
DTLS	Datagram Transport Layer Security
IBC	Identity-Based Cryptography
GW	In-vehicle Gateway
CS	Cloud Server
$ID_{GW}$	Identity string of gateway in vehicle
$IF_{num}$	Nthnetwork adapter
$IP_{GW}^{IF_{num}}$	IP address of network adapter num
$s_{GW}$	Primary private key of GW security domain
$S_{CS}^{GW}$	Registration credential submitted by the GW to the CS
$S_{GW}^{IF_{num}}$	Network adapter private key of network adapter num
$K_{GW}^{IF_{num}}$	The shared key material of network adapter num
$seed^{IF_{num}}$	The number of IP changes of network adapter num
$k^{IF_{num}}$	The shared key of network adapter num
e(,)	Bilinear mapping function
H()	Collision-resistant hash function for character to elliptic curve mapping
||	Concatenation operation
PRF	Pseudorandom function

**Table 3 sensors-22-08623-t003:** Computational cost for cryptographic primitives.

Operations	Time Cost (ms)
pairing initialization $T_{ini}$	2.302
convert string to point on elliptic curves $T_{convert}$	2.067
scalar multiplication on elliptic curves $T_{multi}$	0.921
apply a bilinear map $T_{map}$	0.529
HMAC for authentication header $T_{auth}$	0.007

**Table 4 sensors-22-08623-t004:** Latency reduction percentage.

	Loss Rate	0%	5%	10%	15%
Mode					
Proposed/interactive		77.8%	88.5%	92.1%	94.0%
Proposed/SFVEC		46.8%	77.7%	84.4%	87.6%
Transmit/interactive		73.9%	85.9%	89.5%	91.6%
Notification/interactive		49.3%	28.3%	20.5%	16.4%

**Table 5 sensors-22-08623-t005:** Link configurations.

Parameters	Path A	Path B	Path C
PLR $\left(\%\right)$	1.5–2	2–2.5	1–1.5
bandwidth	4 M	4 M	4 M
delay	45–55 ms	65–75 ms	55–65 ms

## Data Availability

Not applicable.

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
