# Peer review of "A Data-Driven Noninteractive Authentication Scheme for the Internet of Vehicles in Mobile Heterogeneous Networks"

_sensors, 2022, doi:10.3390/s22228623_

Round 1
Reviewer 1 Report
A Data-driven NonInteractive Authentication Scheme for the Internet of Vehicles in Mobile Heterogeneous Networks
The authors proposed a data-driven noninteractive authentication scheme, a lightweight, stateless scheme supporting mobility multi-homing to meet the lightweight data security requirements of the IoV scenario.
My comments are as follows:
The contribution of the proposed scheme is not discussed clearly. How this scheme works to secure the entire system?
There are several abbreviations used without proper definition at first occurrence, i.e., IKE, HIP, DTLS
Figure 3 is not cited in the text, check all other figures also for the same. Fig 3 needs to be explained in the text.
Literature review should be updated by adding some more related approaches, i.e.,
Yu, S., Lee, J., Park, K., Das, A.K. and Park, Y., 2020. IoV-SMAP: Secure and efficient message authentication protocol for IoV in smart city environment. IEEE Access, 8, pp.167875-167886.
Usman, Muhammad, Rashid Amin, Hamza Aldabbas, and Bader Alouffi. "Lightweight challenge-response authentication in SDN-based UAVs using elliptic curve cryptography." Electronics 11, no. 7 (2022): 1026.
Wu, Tsu-Yang, Xinglan Guo, Yeh-Cheng Chen, Saru Kumari, and Chien-Ming Chen. "SGXAP: SGX-Based Authentication Protocol in IoV-Enabled Fog Computing." Symmetry 14, no. 7 (2022): 1393.
Captions of some figures must explain the entire figure briefly.
In the proposed scheme, how the security keys can be protected as an intruder can make changes during the communication process.
A complete flow diagram with an explanation should be added.
If there is a malicious GW, it can easily spoof the key information to attack the entire system.
The simulation environment and parameters are discussed in the paper.
What is the computation overhead on the system?
There are several English and grammar errors, i.e., “This study designed a noninteractive authentication scheme that is a network state 458 context-aware authentication scheme for mobile and multihoming in-vehicle gateways 459 operating in wireless heterogeneous network scenarios.”
The conclusion should discuss the insights and improvements of the proposed system.
If possible some realistic parameters should be used for evaluation of the system.
Reviewer 2 Report
The research proposed a noninteractive authentication scheme and aimed to reduce delay in IOV and bring higher bandwidth aggregation efficiency;
The proposed novel method is interesting, and the paper is generally well-written. However, there are some major and minor issues as below:
Major issues:
- The author in the literature discussed issues related to authentications only; the study should be rich with references in the existing research on the above points to prove the study's necessity.
- Algorithms GW and CS need to be elaborated
- The evaluation process was done based on four indicators; the author doesn't explain the importance of those indicators and why it is necessary for his study.
-The result of the experiment should be compared with other studies.
Minor issues:
-The author explained all terms related to the study well and in a distinctive format; I suggest making a table of abbreviations.
-The figure of the proposed scheme and system architecture is not introduced.
- Figures (1,2,3, and 5) are primitively drawn and are supposed to be more professional, whereas some parts of figure 7 are missing and out of consistency.
-There was no limitation to the study and no future work; it would be better to be planned.
Reviewer 3 Report
The authors considered a noninteractive authentication scheme for the Internet of Vehicles. The problem is important. However, the authors should consider the following comments to improve the paper:
- The analysis of related work was weak. It would be better if the authors could show a gap in previous research on noninteractive authentication schemes.
- The description of the proposed authentication scheme should be presented more clearly. The authors should describe the outline and main steps of the proposed authentication scheme before going into detail.
- Many formulas presented in the paper should be explained more clearly.
- The reader would appreciate it if all notations could be summarized in a table.
- Algorithm 1 and Algorithm 2 should be explained more clearly.
- Figure 7 should be revised.
- The paper should be proofread.
Round 2
Reviewer 1 Report
The captions of some figures, i.e., Fig 4, 7, 9, 10 must be descriptive.
Based on the parameters discussed, how much percentage improvement is observed with respect to existing approaches.
Reviewer 2 Report
The authors have properly replied to my comments. The paper has been improved.
Reviewer 3 Report
The paper has been improved. However, it would be better to present the paper more clearly. Spelling errors and grammatical mistakes in English should be corrected.
